# P^4^UIoT: Pay-Per-Piece Patch Update Delivery for IoT Using Gradual Release

**DOI:** 10.3390/s20072156

**Published:** 2020-04-10

**Authors:** Nachiket Tapas, Yechiav Yitzchak, Francesco Longo, Antonio Puliafito, Asaf Shabtai

**Affiliations:** 1Dipartimento di Ingegneria, Università degli Studi di Messina, Contrada Di Dio (S. Agata), 98166 Messina, Italy; flongo@unime.it (F.L.); apuliafito@unime.it (A.P.); 2Department of Software and Information Systems Engineering, Ben-Gurion University of the Negev, 84105 Be’er Sheva 653, Israel; yechiavil@gmail.com (Y.Y.); shabtaia@bgu.ac.il (A.S.); 3CINI: National Interuniversity Consortium for Informatics, Via Ariosto 25, 00185 Rome, Italy

**Keywords:** Internet of Things, blockchain, patch update, distributed ledger, bitcoin, lightning network, incentivized system

## Abstract

P${}^{4}$UIoT—pay-per-piece patch update delivery for IoT using gradual release—introduces a distributed framework for delivering patch updates to IoT devices. The framework facilitates distribution via peer-to-peer delivery networks and incentivizes the distribution operation. The peer-to-peer delivery network reduces load by delegating the patch distribution to the nodes of the network, thereby protecting against a single point of failure and reducing costs. Distributed file-sharing solutions currently available in the literature are limited to sharing popular files among peers. In contrast, the proposed protocol incentivizes peers to distribute patch updates, which might be relevant only to IoT devices, using a blockchain-based lightning network. A manufacturer/owner named vendor of the IoT device commits a bid on the blockchain, which can be publicly verified by the members of the network. The nodes, called distributors, interested in delivering the patch update, compete among each other to exchange a piece of patch update with cryptocurrency payment. The pay-per-piece payments protocol addresses the problem of misbehavior between IoT devices and distributors as either of them may try to take advantage of the other. The pay-per-piece protocol is a form of a gradual release of a commodity like a patch update, where the commodity can be divided into small pieces and exchanged between the sender and the receiver building trust at each step as the transactions progress into rounds. The permissionless nature of the framework enables the proposal to scale as it incentivizes the participation of individual distributors. Thus, compared to the previous solutions, the proposed framework can scale better without any overhead and with reduced costs. A combination of the Bitcoin lightning network for cryptocurrency incentives with the BitTorrent delivery network is used to present a prototype of the proposed framework. Finally, a financial and scalability evaluation of the proposed framework is presented.

## 1. Introduction

The last decade has seen tremendous growth in the Internet of Things (IoT) services and devices owing to rapid advancements in networking technologies. Gartner, Inc. predicts a 21% growth to 5.8 billion IoT endpoints by 2020 compared to 2019 [1]. The growth is expected to reach 64 billion devices worldwide by 2025 [2], with the predicted market size to reach $520 billion by 2021 [3]. With recent advances in next-generation mobile connection technology 5G, mobile subscriptions are predicted to reach 1.3 billion by 2023 [4]. The ubiquitous nature of IoT devices makes them a primary suspect for attackers. Recent exploits like DDoS attack [5], or ZigBee chain reaction [6] demonstrated in practice point towards weak security posture of many popular IoT devices. Other attacks include break into homes [7,8], compromise local networks [9,10] and smart devices [11]. However, with an increase in focus on the security of IoT devices [7,12,13,14,15], the practice of patching the IoT devices with security updates is a simple solution to protect them from cyber-attacks.

Despite being a basic solution, patching is often ignored or scarcely performed, as observed by the users and manufacturers alike [16]. Narrow profit margins and operational difficulties limit the large-scale patching of IoT devices by the manufacturers. The de-facto client-server-based centralized distribution mechanism, specifically for IoT patch updates, is another cause of concern. The volume of IoT devices and the data generated and consumed by them stresses the ISPs, inter-ISP business relationships, and the Internet backbone. Thus, researchers are focusing on edge computing solutions to limit the information exchange to a single ISP [17,18]. Also, centralized distribution depends heavily on centrally controlled and widely spread cloud service. The centralized control makes the system vulnerable to local outages or natural calamities, as well as exposes it to significant central points of failure [19]. Even spreading the cloud servers to multiple regions still makes the system vulnerable to organizational faults and human errors.

Considering the limitations of existing systems, a distributed P2P content delivery network holds promise. In particular, they can be explored to optimize patch delivery to IoT devices. For example, consider the file-sharing networks like Gnutella [20], IPFS [21], and BitTorrent [22], which became popular in the last decade. As a case in point, large organizations and enterprises like Microsoft (for Windows 10 updates [23]), Twitter (to speed up servers deployment [24]), Spotify (to reduced its hosting costs [25]), or Amazon [26], are exploring P2P distribution. Unfortunately, such attempts of peer-to-peer distribution are limited to a few organizations acting as peers on the Internet. To truly reach its potential, the distribution needs to be more inclusive and, thus, incentivized. Unfortunately, such attempts are severely limited to independent Internet peers.

Also, such systems suffer from a fundamental problem: limited availability in case of unpopular files like patch update. For example, in the case of IoT patch update distribution, the IoT devices are the only parties interested in the update uploaded by a vendor. Thus, other peers of the network will not be interested in downloading and sharing it in the absence of any incentive. Even the IoT devices will not be able to share the files due to limited resources available.

The authors in [27,28] propose a blockchain-based IoT patch distribution to improve accountability and availability. However, in the absence of incentives, the network did not scale beyond the nodes controlled by the manufacturers. Lee et al. [29] propose a cryptocurrency incentive mechanism for encouraging a network of distributor networks to deliver patches to destination IoT devices. Leiba et al. [30] propose a similar approach to [29], but with an efficient distribution mechanism. Both proposals enable a fair exchange of authenticated software updates and cryptocurrency payments. However, an on-chain payment solution suffers from several problems. (i) Costs: Each transaction on blockchain costs transaction fees in addition to incentives being transferred. For example, the Bitcoin transaction fee is reaching around 60 cents. https://bitinfocharts.com/comparison/bitcoin-transactionfees.html. IoTPatchPool [30] analyzed per-device fees to be around 10 cents. (ii) Latency: The delay caused due to the required number of block confirmation prevents the solution from scaling. For example, an average block creation time in Bitcoin is ten minutes, and it needs at least six blocks to confirm a transaction. Thus, a single update may take around one hour. (iii) Throughput: Due to the latency delay, the device update is limited by an upper bound within a given time frame. (iv) Privacy: Being a public ledger that can be audited by anyone, blockchain lacks privacy. An attacker can learn critical information like the number of devices handled by the vendor, how many devices got patched, the cost of patching the devices, etc.

In this work, we present P${}^{4}$UIoT, a pay-per-piece patch update for IoT software updates using a gradual release, an incentivized distributed delivery network based on Bitcoin’s Lightning Network [31]. The transaction fee is reduced, as a single transaction with a commitment is mined on the blockchain. This commitment is known as a channel. Once confirmed, further transactions happen purely peer-to-peer thus, improving the throughput of the system. Also, latency is reduced to network latency. Privacy is preserved thanks to an underlying onion-routing scheme and having just the payments’ involved parties aware of a payment made, and its terms. The concept of gradual release refers to dividing and exchanging the commodity between the sender and receiver in rounds wherein the trust between participants grows with each round. We formally analyzed the proposed approach using TLA+ [32] formal modelling language. Also, we evaluated the correctness of the formal specification. We implemented a prototype of the proposed framework on the Bitcoin’s lightning network to evaluate its performance. We performed experiments to evaluate the distribution latency and cost metrics in various test scenarios. Finally, we presented the advantage of the proposed framework by comparing it with existing work in the literature.

The remainder of the paper is structured as follows. After discussing related work in Section 2, we present the threat model and motivation in Section 3 and list the preliminaries in Section 4. Section 5 and Section 6 describe the proposed framework and blockchain implementation of the flow for validating a design file, respectively. Section 7 presents a formal specification of the proposed solution using TLA+ formal language. In Section 8 and Section 9, we discuss the correctness of the model, possible real-world fraud scenarios, performance and financial statistics, and future work.

## 2. Related Work

In this section, we present the literature from the decentralized storage networks and IoT devices patching domain. The current work spans across these domains.

### 2.1. General Architectures for IoT Patching

The traditional delivery network for patch updates used to be host-centric, where the vendor used to provision the updates to the clients. However, with the advent of the Internet of Things, there is a massive explosion in number devices, making this solution challenging to scale. The complex infrastructure requirements and the personnel to manage it, increased the maintenance cost for the software providers. Liu et al. [33] and Zhen-hai and Yong-zhi [34] are some of the studies that outlined the challenges around availability and reliability by outsourcing infrastructure maintenance to IaaS and a Maven-based solution, respectively. However, the proposed solutions were limited by the number of IoT devices. Yu et al. [35] tried to address the IoT devices’ security via traditional solutions like antivirus and software patches. With the scale and diversity of IoT devices, there was a need for a paradigm shift in security solutions. The authors of [36] propose a solution based on device diversity for a particular scenario of vehicular networks. Similar solutions were proposed in [37,38] with a non-trustworthy assumption. However, the problem of scalability, which is central to IoT devices, is not addressed.

The authors of [27] proposed a blockchain and peer-to-peer file-sharing-based patch distribution system, which is secure, highly available, and allows versioning of updates. A similar idea was proposed in [28] by Boudguiga et al. with the addition of a trusted entity responsible for verifying the update. Both the proposals acknowledge the complexity of sharing a patch update on the blockchain and, thus, propose an off-chain solution for distribution. However, in the absence of an incentive mechanism for the distributor nodes to distribute the patch updates, the solutions are limited in scale and marginally better than the centralized vendor network.

Lee [29] proposed a blockchain-based solution for patch distribution with the incentive for delivery. A unique receipt is generated for each delivered patch, and only the distributor successfully providing the patch can claim the reward. This is achieved by registering the identity of the distributor (encryption keys), which is then used to preserve the integrity of the patch update. However, the uniqueness property forces the vendor to distribute the package to every distributor to have fairness in the system. Thus, the system has similar bandwidth requirements as those of a centralized solution. Furthermore, the creation of blockchain transactions for each patch delivery by a distributor is a costly and time-consuming process. Also, the solution requires the IoT device to maintain a digital wallet to pay for the update, which introduces additional vulnerabilities and attack vectors. Finally, the wallets need to be provisioned, causing additional transactions on the blockchain. IoTPatchPool [30] presents another blockchain-based patch delivery network with an incentive for every successful delivery. The patch distribution is done via a distributed storage network like BitTorrent, while the blockchain network handles the security guarantees and incentives. The security guarantees are encoded in the smart contract and are stored on the blockchain for public verifiability. The distributors claim the reward from the smart contract by presenting a proof-of-distribution for each successful delivery. The solution, though promising, does not scale well as the contract creation, committing the proof-of-distribution, and revealing the secret for claiming the reward create transactions on the blockchain creating roughly one hour of latency per transaction. IOTA [39] is another approach supporting micro-transactions, making it promising for patching IoT devices. However, initially designed for service exchange between IoT devices, the incentive mechanism of IOTA requires initiation from IoT devices, making it unsuitable for security patching use cases.

We propose a solution based on blockchain focusing on incentivizing the security patch delivery, which is delegated to the non-vendor nodes. Also, the trust issue between the vendor and the distributors is addressed for rapid scaling.

### 2.2. Decentralized Storage Networks

Decentralized storage networks (DSN) distribute digital content among the participants via peer-to-peer networking. Also known as P2P file-sharing systems, the underlying architecture resembles that of blockchain, making it a suitable candidate for a blockchain-based distribution scheme. Incentivization is necessary in the case of P2P, as peers tend to free-ride, affecting the performance and availability of the network [40,41]. Thus, incentivization must be a part of any solution based on DSN. BitTorrent’s choking algorithm [22] is an incentivized “quid pro quo” exchange scheme in which an uploader can eliminate a non-participating peer. The scheme is promising but is limited in case of unpopular content like patch updates. This affects the long-term availability of the content [42].

Incentivization makes more sense in IoT use case where the 3rd party distributors are expected to share unpopular content necessary to secure the IoT devices. It is reasonable to assume that the IoT devices, being resource constraint, will not be able to share the update among themselves. Given the untrusted and incentivization oriented situation, blockchain seems to be the solution. It allows a trustless exchange of service and cryptocurrency among its users. Siacoin [43], Storj [44], Swarm [45] and Filecoin [46] are some examples of blockchain-based DSN solutions.

A summary of the related works is presented in Table 1. The *Incentives* refers to the cryptocurrency used to encourage distribution. *Trust* represents the fairness in an exchange between the patch update and the incentive. *Scaling* refers to the increase in distribution latency with the increase in the number of devices. A solution scales well if the distribution latency increases gradually with an increase in the number of devices. *Solution* refers to the state of the presented solution, where D stands for design, I stands for implementation, and S stands for simulation.

All the above solutions assume trustless exchange between participating entities and that payment channel supports it. The authors in [31,47] propose such channels capable of handling large volumes with low transaction fees. However, they also impose some restrictions on the participating entities, which can be challenging in an IoT use case. (i) To be financially economical, the amount of data transfer needs to cover the cost required to fund the devices, create a channel, and close a channel for a device. (ii) The device is expected to hold hot wallets and lock funds. (iii) The disputes need to be actively resolved by the participants by observing the ledger either by themselves or via a trusted 3rd party. (iv) Since any file is distributed in parts, a malicious user might collect different parts from different peers without any incentive.

The price of negotiation is another complex aspect that is usually left to the consumers. The assumption of a single vendor handling the distribution can simplify the management of pricing.

## 3. Security Model

In this section, we present a threat model and the assumptions related to the participants. Based on the model, we outline the security goals of the system design.

### 3.1. Threat Model

The proposed architecture consists of three kinds of participants: a *vendor* who wants to distribute the update, a *distributor* who shares a part or whole of an update in exchange for an incentive, and an *IoT device* that needs and pays for the patch. We assume that the vendor, the distributor, and the IoT device can be uniquely identified and can securely communicate with each other via standard protocols and credentials like https. The IoT device is identified by its public key, which the vendor pushes into the device during the manufacturing process. A more robust mechanism for device identity like SSI [48] can be used to secure the system further and is beyond the scope of this paper. Also, we assume that the distributor/s and the IoT devices are aware of the public key of the vendor/s, and the identity of IoT devices is available with the vendor. These assumptions can be achieved during the manufacturing of an IoT device or configured by an admin. We also assume that the vendor can be trusted to work in favor of the network and does not accept the hash of a malicious file, and IoT devices trust the patch update sent by the vendor.

The IoT devices and the distributors can assume malicious intent towards each other and can try to claim updates/incentives.

### 3.2. Adversarial Model

We assume that an adversary has full control over the communication medium and can alter any communication with the blockchain or any participating entity [49]. We also assume that the adversary can take a passive as well as an active role and perform actions varying from listening on the network packets to injecting, replaying, or filtering any message.

### 3.3. Security Goals

Based on the adversarial and trust profile, we outline the security goals that our proposal should satisfy: (i) an IoT device pays for a patch before receiving it; at the same time, a distributor delivering a patch is paid. (ii) the IoT device can verify the integrity and the origin of the patch in the presence of malicious actors. (iii) within a given time frame allocated by a vendor, an IoT device will be able to patch itself.

## 4. Preliminary

In this section, we briefly introduce the participating entities of the system. We also describe the requirements related to the entities. The general architecture is based on the work presented in [30].

### 4.1. Decentralized Storage Network (DSN)

The peer-to-peer or decentralized storage network should be available to all the participating entities. The peer discovery is based on the distributed hash table (DHT), allowing access to all peers without imposing any substantial requirements. Based on DHT, each peer is aware of all the peers holding a part of the data. The proposed system distributes patch updates using BitTorrent DSN based on the Kademlia DHT algorithm [50]. The algorithm ensures efficient lookup of the order of log(N) where *N* stands for the number of nodes.

### 4.2. Blockchain Network

The blockchain network is considered to be permissionless and must support cryptocurrency intrinsically. Thus, any participating entity can read and verify transactions on the blockchain. Bitcoin [51] is the widely adopted permissionless blockchain. All the participating entities either hold a node or are connected to a trusted node on the blockchain network.

### 4.3. Vendor Nodes

The IoT device manufacturers own host machines that are part of both the blockchain network and the distributed storage network. These machines are defined as vendor nodes in the framework. Since the vendor node needs only to verify the operations on the blockchain, the vendor can maintain a light node on the blockchain network.

The following information must be maintained by each vendor node:a pair of public and private keys $\left(pk_{i},sk_{i}\right)$.a collection of IoT device identities. This can be achieved by configuring a private key to each device and accumulating the corresponding public keys.

### 4.4. IoT Nodes

The framework defines the IoT devices that need to be updated by the vendor node as IoT nodes. They participate in both the blockchain network and the distributed storage network. However, considering the resource constraints of an IoT device, the blockchain network can be joined by the IoT nodes either as a light client or delegating the trust on a 3rd party full node on the network. Also, the distributed storage network does not limit the operation of an IoT device. The IoT device assumes the role of a consumer on the DSN network and, thus, does not share any file on the network. Finally, the internal routing table maintained in the form of DHT for peer discovery requires a limited amount of memory.

The following information must be maintained by each IoT node:a pair of public and private keys $\left(pk_{j},sk_{j}\right)$;public key of its vendor node, $pk_{i}$.
The vendors’ public key can be configured on the IoT device during the manufacturing by the vendor.

### 4.5. Distributor Nodes

The independent peers that participate in the distribution of the patch update are defined as Distributor nodes. The distributor nodes, like others, need to participate in both the blockchain network and distributed storage network. Similar to the IoT node, the distributor must be able to route transactions on the network. However, the distributor is not limited in resources and, thus, can run a full blockchain node.

## 5. Pay-Per-Piece Patch Update

This section describes the P${}^{4}$UIoT—pay-per-piece patch update framework, its building blocks, and the design of the pay-per-piece digital exchange protocol, which enables it to achieve an open incentivized system. Pay-per-piece is a general exchange protocol that includes two or more parties that are interacting in a trustless exchange of goods. The core concept is the fact that each party will reduce its risk exposure by dividing the patch by a particular factor. The trust is built over time with multiple cycles of exchange of the patch piece and incentive.

The proposal will try to address a few of the shortcomings of the former frameworks, which are as follows:Patch size limitations: the IoTPatchPool [30] framework relies on the ZKCP protocol, which is limited in its current implementation for 55 kB.Network overhead: in addition to the patch update itself, the former framework requires the sending of additional data in the form of ZKsnarks proving and verification keys; in some cases, it is a hundred folds bigger than the patch file itself.Demands for IoT devices: in the ZCKP protocol, the IoT devices need to verify the ZKsnarks statement, and also, there is a need to decrypt the patch binaries once the decryption key is revealed. These two operations are having additional demands on the device storage memory and processing power.Costs: the former network is a first layer solution on the blockchain network. To reduce costs, new methods of offering payment to the distributors should be used.

The proposed framework consists of two networks, a lightning network over a blockchain network used to enable a transparent exchange of patch update and cryptocurrency incentive in the form of binding bid, and a decentralized storage network (DSN) which allows a peer-to-peer sharing of a patch update. Icons made by https://www.flaticon.com/authors/freepik.

The participating entities of the network, namely Vendors, Distributors, and IoT devices, use the two networks to follow the protocol proposed in the framework. Figure 1 represents an overall architecture of the proposed solution, focused for the sake of clarity on a smart home use case. The delivery life-cycle is initiated by a vendor that needs to distribute a new patch update to the deployed IoT nodes. The vendor invites the distributors interested in the task of providing the update to the IoT nodes by committing a bid on the blockchain, which will be gradually released with the proof-of-distribution. The IoT nodes will release the proof-of-distribution to the distributor upon receiving a piece of patch update, who can then exchange it for the cryptocurrency incentive. Also, for a limited duration, the vendor acts as a source of distribution of patch update for the distributors and IoT nodes alike. Once done, the distributors take over the process, seeding the pieces downloaded from the vendor via the DSN.The IoT nodes, upon notification of a patch release, download the pieces of update from distributors via the DSN and release the corresponding proof-of-distribution.

Leading from the previous work on the IoTPatchPool [30], the proposal adds additional methods to achieve a trustless protocol for IoT patch distribution. The desired framework should accompany all the former framework properties and address a few of the shortcomings listed earlier. The pay-per-piece protocol requires an exchange for signature between the distributor and the IoT during the file exchange. As a security parameter for the system, the smaller pieces and hence larger count of pieces is preferable since it minimizes the initial risk in the fair exchange. On the other hand, a considerable number of signatures means higher fees paid in the form of the transaction cost. One method to achieve a lower number of signatures sent to the smart contract is signature aggregation. The ability to compact multiple signatures into one signature will lead to smaller fees paid to the miners.

The following are the two approaches integrated into the Bitcoin protocol to reduce the signature size.

Schnorr Signatures were introduced to the Bitcoin network in [52]. Given *n* signatures on *n* distinct messages from *n* distinct users, it is possible to aggregate all these signatures into a single short signature. The tests conducted show a significant speedup. The ratio between the time it takes to verify *n* signatures individually and to verify a batch of *n* signatures goes up logarithmically with the number of signatures or, in other words: the total time to verify *n* signatures grows with O(n/log(n)).ECDSA Elliptic Curve Digital Signature Algorithm [53] is one of the main building blocks of the Bitcoin network, and it is used for verifying all the transactions. Reviewing all the possibilities, the ECDSA alternative was chosen for its low costs and lack of demand for preliminary online setup between signers. It is important to notice that choosing ECDSA will allow this framework to be scalable as the blockchain ledger itself is making additional improvements in scalability to be applied to this framework.

The Schnorr Signatures and ECDSA together can improve the verification time with signature aggregation and key size reduction [54]. This is particularly advantageous in the case of resource-constrained devices.

### 5.1 Protocol

In this section, we outline the proposed protocol for sending patch update to a set of IoT devices $d_{1},\ldots ,d_{N}$ manufactured by a vendor V. Let us denote the patch update by U. The proposal consists of five stages: initial setup, bid commitment, initiating patch update seeding, exchanging patch piece for a pay-per-piece confirmation, and a reward claim. Detailed sequence diagram presented in Figure 2.

#### 5.1.1. Initial Setup

The proposed framework uses connection establishment as an initial step for any patch update procedure. The connection links the vendor to the community of distributors who would like to participate in future IoT patch update procedure. The distributor would listen to an event emitted by the vendor to receive the newly published patch and initiate the process of a patch update.

#### 5.1.2. Bid Commitment

The vendor V sends a transaction to the blockchain committing the incentive $I_{i}$ for each of the IoT devices $d_{i}$. Once committed on the blockchain, a peer-to-peer payment channel is established between each distributor $k_{i}$ and each of the IoT nodes $d_{i}$.

#### 5.1.3. Initiating Patch Update Seeding

The vendor V performs the following steps on receiving an update U:Computes the hash of patch file $U_{t}:=H\left(U\right)$Sets N as the total count of IoTs $d_{i}$Sets M as the total count of piecesPrepares the payload as: $P:=(U,sign_{v_{i}}\left\{U_{t}\right\},d_{i})$, where $d_{i}$ is represented as an ordered list of the IoT devices public keysComputes the hash of the payload $P_{t}:=H\left(P\right)$Sets $\Delta_{LifeTime}$ as sufficient time for vendor V to seed the patchEmits an event announcing the availability of patch update.
Both the distributor nodes $k_{i}$’s and IoT devices $d_{i}$’s are listening to the events emitted by the vendor V known by his public key $pk_{i}$. Once the event is received, the following set of operations are performed:Corresponding to the hash $P_{t}$, the distributor requests vendor V for the payload via the DSN network.The vendor V begins the seeding of the payload P using the DSN network. The vendor seeds the payload P for a reasonable time within which some distributor $k_{i}$ can receive the update.After completely downloading the package P, $k_{i}$ verifies that:
the patch is published by the vendor.the integrity of the patch is maintained throughout the communication.$k_{i}$ then announces via DHT to its peers as a possible source of the patch update.

#### 5.1.4. Exchange

In this step, the IoT device, and distributor communicate to exchange the patch update and proof-of-distribution. Initially, a verification of the IoT device is conducted to understand if it is a member of the vendor-approved list $d_{i}$. The detailed exchange is listed below:The patch update available event is received by the IoT device $d_{i}$, and the device verifies the sender of the event.The device $d_{i}$ searches for distributor $k_{i}$ possessing of the patch update via the DHT peer discovery. Once found, the device sends the download request to the distributor $k_{i}$.$k_{i}$ sends a random nonce *c*, a challenge, to $d_{i}$ to sign on.$d_{i}$:
Computes $sign^{d_{i}}\left\{c\right\}:=Sign\left(sk^{d_{i}},c\right)$.Sends the tuple $\left(pk^{d_{i}},sign^{d_{i}}\left\{c\right\}\right)$ to *k*.In this stage, the IoT device can start downloading pieces from any free providing distributors (i.e., the vendor node).$d_{i}$:
Verifies that $pk^{d_{i}}\in D_{i}$Verifies that $VerifySig(pk^{d_{i}},sign^{d_{i}}\left\{c\right\},c)=1$Piece by piece exchange:
$d_{i}$ sends the requested *j* part of the patch file$k_{i}$ concats $M_{j}$ = $concact(U_{t},pk^{d_{i}},j)$ and sends to $d_{i}$$k_{i}$ sends part to $d_{i}$$k_{i}$ chokes further transfer till the payment for the part is received.

#### 5.1.5. Reward Claiming

The reward is claimed by the distributor of the piece sent as follows:$d_{i}$:
once the piece is received, $d_{i}$ sends the payment of the piece to $k_{i}$.once $k_{i}$ received the payment, it sends the next piece and waits for payment.

## 6. Implementation

The P${}^{4}$UIoT implementation proves the capability of such a framework to function in the multi-party environment. Figure 3 refers to the technology stack used for the implementation. Raspberry pies are used as a fleet of IoT devices that request patch from the distributors, which in return communicate with the lightning network to reclaim their reward. The implementation of the proposal needs an evaluation of the new features of the framework. One key question is *the latency of the blockchain*. Since the proposal is based on the lightning network, once the vendor sends the commitment to the blockchain, the payment is processed in real time between the distributor and IoT device. Another key question is *the costs*. The commitment is the only transaction sent by the vendor on the blockchain. Assuming multiple usages, the transaction cost is negligible. To answer these questions, the implementation is based on the lightning network and deployed over the Bitcoin’s regtest. The Bitcoin and lightning network protocols are used as is, and no changes has been made to the original protocols. The transaction prices are aligned with the mainnet, and the block-time is similar to using PoW as the consensus algorithm. Finally, all attacker scenarios were tested, among reward interception, malformed channel, Denial-of-service (DoS) redeem with false message, exhausting resources, blockchain-related attacks, compromising a firmware’s integrity, software downgrade attack, greedy distributor, and greedy IoT node.

## 7. Formal Specification

In this section, we present a formal specification of the proposed approach using TLA+ abstract language [32] for model verification. Using TLA+ definitions, we define the system interaction of P${}^{4}$UIoT and verify the correct operation of the system. Finally, using the formal specification, we check for the correctness of the proposed solution using TLC model checking tool in the next section.

### 7.1. State Transitions

There are three participants in the proposed approach, the vendor, the distributor, and the IoT. The payment (incentive) system is based on the lightning network running on top of the Bitcoin network. The Bitcoin network and the vendor is an offline entity, while the rest of the entities are online. The state transition diagram in Figure 4 presents the system behavior. The vendor generates the package to be distributed, as well as establishes a payment channel with the IoT devices. The package preparation and channel establishment are done offline. The distributors and the IoT nodes have their own states to process the patch update. The communication and payment between the distributors and the IoT nodes happen through channels.

The TLA+ action formula is used to represent individual states of the distributors and IoT nodes, and their transitions. We represent the unchanged variables as primed variables (indicated by the UNCHANGED keyword in the action formula). Each step consists of the following:the entry condition, andthe variable changes
Detailed specifications on TLA+ can be found in [32,55].

We begin by initializing each variable with a default value. The variables InitVendor, InitDistributor, and InitIoT are initial values of the variables for Vendor, Distributors, and IoTs, respectively. The channel and payment_channel are used for message and payment exchange between distributors and IoTs. Init is defined as:$$\begin{array}{cc} Init & ≜\\ \; & \wedge InitVendor\\ \; & \wedge InitDistributor\\ \; & \wedge InitIoT\\ \; & \wedge balance=[c\in Distributor\cup IoT\mapsto InitBalance]\\ \; & \wedge channel=\{\} \end{array}$$

The state change for the proposed model is defined as $Init\wedge *\left[\;Next\right]{\;}_{vars}\wedge WF\_vars\left(Next\right)$ in TLA+ specifications where the WF_vars is for fairness. The possible transitions from the initial state are described in the following equation: $$\begin{array}{cc} Next & ≜\\ \; & \vee Seeding\\ \; & \vee \exists \;k\in Distributors\;:\\ \; & \;\;\;\;\vee Receive(k)\vee Delivery(k)\vee PaymentRequest(k)\\ \; & \;\;\;\;\vee VerifyPayment(k)\vee TryAnother(k)\\ \; & \vee \exists \;d\in IoT\;:\\ \; & \;\;\;\;\vee Verification(d)\vee Payment(d)\vee RejectPiece(d)\\ \; & \vee (*\;\mathrm{Disjunct}\;\mathrm{to}\;\mathrm{prevent}\;\mathrm{deadlock}\;\mathrm{on}\;\mathrm{termination}\;*)\\ \; & \;\;\;\;((\forall \;q\in IoT:dState[q]=I\_Final)\\ \; & \;\;\;\;\;\;\;\;\wedge Terminating)\\ Spec & ≜\\ \; & Init\wedge *\left[\;Next\right]{\;}_{vars}\wedge WF\_vars\left(Next\right) \end{array}$$

We consider the case of patch delivery by Vendor to multiple IoT nodes with the help of multiple Distributors. In the initial step, the vendor is responsible for preparing the package to be distributed and metadata related to the distribution, such as the distributors and IoT nodes involved in the process (identified by their public keys), mechanism to ensure integrity of the package, etc. The blockchain is not actively involved in the system interactions as the scope is limited to creation of lightning channels for the payment. The channel creation is considered to be part of pre-process and is thus not part of the state transition. Termination describes the conditions in which P${}^{4}$UIoT is considered to be stable i.e., the patch has been distributed to the IoT devices successfully and the Distributors have been paid.

### 7.2. Vendors

The Vendor is responsible for preparation of the package to be distributed to the IoT devices. We define the Seeding as a combination of TLA+ actions. For the sake of simplicity, we use external oracles to define core cryptographic primitives and provide these values externally during the verification process. As per the proposal, the keys of the communicating entities are registered with each other. For example, the Vendor’s public key is known to the Distributors and the IoT devices. EPAYLOAD is the data structure used to provide payload. regKey holds the public keys of the target nodes which act as their identities on the network. torrentinfohash and filehash are used to verify the integrity of the torrent and file content. Finally, the patch update availability is posted publicly to all entities and the patch update is seeded by the torrent network.
$$\begin{array}{cc} Seeding & ≜\\ \; & \exists \;pu\in regKey:\\ \; & LET\\ \; & \;\;\;\;\;\;payload≜EPAYLOAD[1]\\ \; & \;\;\;\;\;\;torrentinfohash≜EPAYLOAD[2]\\ \; & \;\;\;\;\;\;filehash≜EPAYLOAD[3]\\ \; & \;\;\;\;\;\;signature≜EPAYLOAD[4]\\ \; & \;\;\;\;\;\;target\_node≜NODE\_TYPE\\ \; & IN\\ \; & \;\;\;\;\;\;\wedge regKey^{\prime }=regKey\backslash pu\\ \; & \;\;\;\;\;\;\wedge Post(\langle pu,payload,target\_node,torrentinfohash,filehash,signature\rangle )\\ \; & \;\;\;\;\;\;\wedge pTime^{\prime }=pTime+1\\ \; & \;\;\;\;\;\;\wedge UNCHANGED\langle distributorVars,iotVars,commVars\rangle \end{array}$$

### 7.3. Distributors

A Distributor transits through different states based on the enabling conditions. The Distributor begins with a D_Waiting and then transits through D_Delivering, D_VerificationWaiting, D_ReceiptWaiting, and D_Final. The actions associated with a Distributor on the above defined states.

*Receive:* Initially, a Distributor is waiting for a patch update from the Vendor. The Distributor is said to be in D_Waiting state. Once the Distributor receives the patch, it performs verification on the integrity of the received file and existence of duplicate package. Once the integrity and uniqueness is verified, the package is stored in a buffer and the Distributor makes a transition to D_Delivering.
$$\begin{array}{cc} Receive(k) & ≜\\ \; & \wedge kState[k]=D\_Waiting\\ \; & \wedge Len(patchPool)>0\\ \; & \wedge LETpatch==Head(patchPool)\\ \; & \;\;\;\;\;\;IN\\ \; & \;\;\;\;\;\;VerifyPatch(patch[2],patch[4],patch[5],patch[6])\\ \; & \wedge kBuffer^{\prime }=\left[kBuffer\;\;EXCEPT\;\;!\left[k\right]=Head\left(patchPool\right)\right]\\ \; & \wedge kState^{\prime }=\left[kState\;\;EXCEPT\;\;!\left[k\right]=D\_Delivering\right]\\ \; & \wedge patchPool^{\prime }=Tail\left(patchPool\right)\\ \; & \wedge UNCHANGED\;\langle regKey,pTime,kIndex,kReceipt,iotVars,commVars\rangle \end{array}$$

The first few lines are entry level conditions namely the current state is D_Waiting; the patchPool contains a package to be distributed, and the package integrity is verified. Once satisfied, the package is stored in buffer and the state is changed to D_Delivering. The variables related to IoT and lightning channel remain unchanged.

*Delivery:* The Distributor in D_Delivery state uses the IoT device identities to find the target devices and delivers the patch to the IoT devices piecewise. The Distributor uses the Send message to send a piece of the package with source, piece index, price of a piece, and lightning receipt. After the transmission, the Distributor chokes the next piece and waits for the receipt of payment for the piece.
$$\begin{array}{cc} Delivery(k) & ≜\\ \; & \wedge kState[k]=D\_Delivering\\ \; & \wedge \exists \;d\in IoT:\\ \; & \;\;\;\;\;\;\wedge dState[d]=I\_Waiting\\ \; & \;\;\;\;\;\;\wedge dIndex[d]=kIndex[k]\\ \; & \;\;\;\;\;\;\wedge LET\\ \; & \;\;\;\;\;\;\;\;\;\;\;\;m\;≜kBuffer[k]\\ \; & \;\;\;\;\;\;\;\;\;\;\;\;pieceindex\;≜kIndex[k]\\ \; & \;\;\;\;\;\;IN\\ \; & \;\;\;\;\;\;\;\;\;\;\;\;Send([src\mapsto k,dst\mapsto d,data\mapsto \langle m[2][pieceindex]\rangle ])\\ \; & \wedge kState^{\prime }=\left[kState\;EXCEPT\;!\left[k\right]=D\_VerificationWaiting\right]\\ \; & \wedge UNCHANGED\;\langle vendorVars,kBuffer,kReceipt,kIndex,iotVars,balance\rangle \end{array}$$

The Distributor transits to D_VerificationWaiting once the piece id delivered to the IoT device. It then waits for the response from the IoT device.

*PaymentRequest:* Once the Distributor receives a successful verification message from the IoT device (M_OK), the Distributor transits to Payment_Waiting state. The Distributor waits for the IoT device to make payment for the piece sent.
$$\begin{array}{cc} PaymentRequest(k) & ≜\\ \; & \exists \;c\in channel:\\ \; & \;\;\;\;\;\;\wedge c.dst=k\\ \; & \;\;\;\;\;\;\wedge c.type=M\_OK\\ \; & \;\;\;\;\;\;\wedge kState^{\prime }=\left[kState\;EXCEPT\;!\left[k\right]=D\_PaymentWaiting\right]\\ \; & \;\;\;\;\;\;\wedge LET\\ \; & \;\;\;\;\;\;\;\;\;\;\;\;index≜kIndex[k]\\ \; & \;\;\;\;\;\;\;\;\;\;\;\;price≜PIECEPRICE\\ \; & \;\;\;\;\;\;\;\;\;\;\;\;receipt≜kReceipt[k]\\ \; & \;\;\;\;\;\;IN\\ \; & \;\;\;\;\;\;\;\;\;\;\;\;RecvInadditionSend(c,[src\mapsto k,dst\mapsto c.src,type\mapsto M\_PaymentRequest,\\ \; & \;\;\;\;\;\;\;\;\;\;\;\;data\mapsto \langle k,index,price,receipt\rangle ])\\ \; & \;\;\;\;\;\;\wedge UNCHANGED\;\langle kBuffer,kIndex,kReceipt,vendorVars,iotVars,balance\rangle \end{array}$$

*TryAnother:* In case the received piece fails the verification at the IoT node, the node returns a M_NO message to the Distributor. In this phase, the Distributor goes back to delivering another patch piece.
$$\begin{array}{cc} TryAnother(k) & ≜\\ \; & \exists \;c\in channel:\\ \; & \;\;\;\;\;\;\wedge c.dst=k\\ \; & \;\;\;\;\;\;\wedge c.type=M\_NO\\ \; & \;\;\;\;\;\;\wedge (\vee \;kState[k]=D\_VerificationWaiting\vee \;kState[k]=D\_PaymentWaiting)\\ \; & \;\;\;\;\;\;\wedge kState^{\prime }=\left[kState\;EXCEPT\;!\left[k\right]=D\_Delivering\right]\\ \; & \;\;\;\;\;\;\wedge channel^{\prime }=channel\backslash \left\{c\right\}\\ \; & \;\;\;\;\;\;\wedge UNCHANGED\;\langle kBuffer,vendorVars,iotVars\rangle \end{array}$$

*VerifyPayment:* On successful verification of the piece, the IoT device pays for the piece via lightning channel. The Distributor verifies the status of the payment and on successful verification, sends the next piece. If the Distributor has successfully delivered all the pieces, the Distributor transits to D_Delivering state to service the next IoT device.
$$\begin{array}{cc} VerifyPayment(k) & ≜\\ \; & \exists \;c\in channel:\\ \; & \;\;\;\;\;\;\wedge c.dst=k\\ \; & \;\;\;\;\;\;\wedge c.type=M\_Paid\\ \; & \;\;\;\;\;\;\wedge balance^{\prime }=[balance\;EXCEPT\;!\left[c.dst\right]=balance\left[c.dst\right]+Fee,\\ \; & \;\;\;\;\;\;\;\;\;\;\;![c.src]=balance[c.src]-Fee]\\ \; & \;\;\;\;\;\;\wedge kState[k]=D\_PaymentWaiting\\ \; & \;\;\;\;\;\;\wedge c.data[1]=STATUS\\ \; & \;\;\;\;\;\;\wedge channel^{\prime }=channel\backslash c\\ \; & \;\;\;\;\;\;\wedge IF\;kIndex[k]=NUMPIECES\\ \; & \;\;\;\;\;\;\;\;\;\;\;\;THEN\\ \; & \;\;\;\;\;\;\;\;\;\;\;\;\wedge kIndex^{\prime }=\left[kIndex\;EXCEPT\;!\left[k\right]=1\right]\\ \; & \;\;\;\;\;\;\;\;\;\;\;\;ELSE\\ \; & \;\;\;\;\;\;\;\;\;\;\;\;\wedge kIndex^{\prime }=\left[kIndex\;EXCEPT\;!\left[k\right]=kIndex\left[k\right]+1\right]\\ \; & \;\;\;\;\;\;\wedge kState^{\prime }=\left[kState\;EXCEPT\;!\left[k\right]=D\_Delivering\right]\\ \; & \;\;\;\;\;\;\wedge UNCHANGED\langle kBuffer,kReceipt,vendorVars,iotVars\rangle \end{array}$$

### 7.4. IoT Devices

An IoT node has transition states namely I_Waiting, I_Verifying, I_RequestWaiting, I_Paying, and I_Final. I_Waiting is the initial state and the IoT node waits for the patch piece from the Distributor.

*Verification:* The Verification process in IoT device is initiated by a Distributor. When the Distributor sends a piece to the IoT device, the IoT device validates the integrity of the piece. We defined the function VerifyMeta to validate the received parameters and return TRUE if the validation succeeds. We will be providing the validation values externally to simulate the cryptographic primitives. In addition, the IoT device validates the identity of the Distributor. In case of successful validation, the IoT device sends M_OK to the Distributor and transits to I_RequestWaiting state. In case the validation fails, the IoT device sends M_NO back to the Distributor and remains in the present state.
$$\begin{array}{cc} Verification(d) & ≜\\ \; & \exists \;c\in channel:\\ \; & \;\;\;\;\;\;\wedge c.dst=d\\ \; & \;\;\;\;\;\;\wedge c.type=M\_Package\\ \; & \;\;\;\;\;\;\wedge dState[d]=I\_Waiting\\ \; & \;\;\;\;\;\;\wedge LET\\ \; & \;\;\;\;\;\;\;\;\;\;\;\;recvPayload≜c.data[1]\\ \; & \;\;\;\;\;\;IN\\ \; & \;\;\;\;\;\;\;\;\;\;\;\;IF\;VerifyPiece(recvPayload,dBuffer[d])\\ \; & \;\;\;\;\;\;\;\;\;\;\;\;THEN\\ \; & \;\;\;\;\;\;\;\;\;\;\;\;\;\;\;\;\;\;\wedge dState^{\prime }=\left[dStateEXCEPT\;!\left[d\right]=I\_RequestWaiting\right]\\ \; & \;\;\;\;\;\;\;\;\;\;\;\;\;\;\;\;\;\;\wedge dBuffer^{\prime }=\left[dBuffer\;EXCEPT\;!\left[d\right]=dBuffer\left[d\right]c.data\right]\\ \; & \;\;\;\;\;\;\;\;\;\;\;\;\;\;\;\;\;\;\wedge RecvInadditionSend(c,[src\mapsto d,dst\mapsto c.src,type\mapsto M\_OK,data\mapsto \langle \rangle ])\\ \; & \;\;\;\;\;\;\;\;\;\;\;\;\;\;\;\;\;\;\wedge UNCHANGED\;\langle vendorVars,distributorVars,dIndex,balance\rangle \\ \; & \;\;\;\;\;\;\;\;\;\;\;\;ELSE\\ \; & \;\;\;\;\;\;\;\;\;\;\;\;\;\;\;\;\;\;\wedge RecvInadditionSend(c,[src\mapsto d,dst\mapsto c.src,type\mapsto M\_NO,data\mapsto \langle \rangle ])\\ \; & \;\;\;\;\;\;\;\;\;\;\;\;\;\;\;\;\;\;\wedge UNCHANGED\;\langle vendorVars,distributorVars,iotVars,dIndex,balance\rangle \end{array}$$

*RejectPiece:* The IoT device rejects the piece already available with it and sends a M_NO message back to the Distributor.
$$\begin{array}{cc} RejectPiece(d) & ≜\\ \; & \exists \;c\in channel:\\ \; & \;\;\;\;\;\;\wedge c.dst=d\\ \; & \;\;\;\;\;\;\wedge c.type=M\_Piece\\ \; & \;\;\;\;\;\;\wedge dState[d]\;\#\;I\_Waiting\\ \; & \;\;\;\;\;\;\wedge RecvInadditionSend(c,[src\mapsto d,dst\mapsto c.src,type\mapsto \\ \; & \;\;\;\;\;\;\;\;\;\;\;\;M\_NO,data\mapsto \langle \rangle ])\\ \; & \;\;\;\;\;\;\wedge UNCHANGED\;\langle vendorVars,distributorVars,iotVars,balance\rangle \end{array}$$

*Payment:* In this phase, the IoT device pays for the piece the Distributor sent to it. The lightning receipt is sent with the piece. The IoT devices decodes the receipt and sends the payment via the payment channel. The state of the IoT device depends on the state of the patch. If the IoT device has received the complete patch, it transits to I_Final state else it transits to I_Waiting.
$$\begin{array}{cc} Payment(d) & ≜\\ \; & \exists \;c\in channel:\\ \; & \;\;\;\;\;\;\wedge c.dst=d\\ \; & \;\;\;\;\;\;\wedge c.type=M\_PaymentRequest\\ \; & \;\;\;\;\;\;\wedge dState[d]=I\_RequestWaiting\\ \; & \;\;\;\;\;\;\wedge IF\;dIndex[d]=NUMPIECES\\ \; & \;\;\;\;\;\;\;\;\;\;\;\;THEN\\ \; & \;\;\;\;\;\;\;\;\;\;\;\;\wedge dState^{\prime }=\left[dState\;EXCEPT\;!\left[d\right]=I\_Final\right]\\ \; & \;\;\;\;\;\;\;\;\;\;\;\;ELSE\\ \; & \;\;\;\;\;\;\;\;\;\;\;\;\wedge dState^{\prime }=\left[dState\;EXCEPT\;!\left[d\right]=I\_Waiting\right]\\ \; & \;\;\;\;\;\;\wedge IF\;c.data[2]=dIndex[d]\\ \; & \;\;\;\;\;\;\;\;\;\;\;\;THEN\\ \; & \;\;\;\;\;\;\;\;\;\;\;\;\wedge dIndex^{\prime }=\left[dIndexEXCEPT!\left[d\right]=dIndex\left[d\right]+1\right]\\ \; & \;\;\;\;\;\;\;\;\;\;\;\;\wedge RecvInadditionSend(c,[src\mapsto d,dst\mapsto c.src,type\mapsto M\_Paid,data\mapsto \langle STATUS\rangle ])\\ \; & \;\;\;\;\;\;\;\;\;\;\;\;\wedge UNCHANGED\langle dBuffer,vendorVars,distributorVars,balance\rangle \\ \; & \;\;\;\;\;\;\;\;\;\;\;\;ELSE\\ \; & \;\;\;\;\;\;\;\;\;\;\;\;\wedge RecvInadditionSend(c,[src\mapsto d,dst\mapsto c.src,type\mapsto M\_NO,data\mapsto \langle \rangle ])\\ \; & \;\;\;\;\;\;\;\;\;\;\;\;\wedge UNCHANGED\langle dBuffer,dIndex,vendorVars,distributorVars,balance\rangle \end{array}$$

### 7.5. Terminating

The system terminates once all the pieces of the patch are delivered to the IoT devices and the Distributors are paid for the piece they delivered. The conditions like FaithfulDelivery, AuthenticatedOrigin, and PackageUniqueness formally evaluate the completion of the patch delivery and are presented in detail in Section 8.
$$\begin{array}{cc} Terminating & ≜\\ \; & \wedge SuccessfulDelivery\\ \; & \wedge PatchIntegrity\\ \; & \wedge UNCHANGED\;vars \end{array}$$

## 8. Evaluation

In this section, we evaluate the correctness of the proposed model based on the formal specifications presented in Section 7. Also, based on the security framework, assumptions presented in Section 3, and the formal specification we evaluate the proposed framework against the threats and its conformance to the stated security goals.

### 8.1. Model Checking with Correctness Properties

The TLC model checking tool spans the state space to find critical events, e.g., violation of always true conditions called invariant, and deadlocks. The formal specification presented in Section 7 is evaluated by the tool to expose such conditions. The tool evaluates the deadlock conditions when the system terminates and checks the invariant on each state.

Successful Delivery and Patch Integrity: The *Successful Delivery* verifies that the patch pieces delivered by the Distributor are correct and received by the IoT devices. It can be verified by checking the state of the IoT device. As per the specification, the IoT device successfully receiving the patch ends up in I_Final state. Thus, the following specification checks for successful delivery.
$$\begin{array}{cc} SuccessfulDelivery & ≜\\ \; & \vee Cardinality(\{d\in IoT:dState[d]=\\ \; & \;\;\;\;\;\;I\_Final\})=Cardinality(NODE\_TYPE) \end{array}$$
$$\begin{array}{cc} PatchIntegrity & ≜\\ \; & \forall \;q\in \{d\in IoT:dState[d]=I\_Final\}:\\ \; & \;\;\;\;\;\;dBuffer[q]=EPAYLOAD[1] \end{array}$$

In patch integrity, we validate the content delivered by the Distributor. For all the IoT devices in the final state, we check if the patch delivered by the Distributor is the same as the patch available with the Vendor. Both these conditions are tested on termination.

Incentive Fairness: To evaluate the fairness of the distribution of incentive among the Distributors, we check the following conditions.
$$\begin{array}{cc} TotalBalanceInvariance & ≜\\ \; & Sum(balance,DOMAIN\;balance)=InitBalance\;*\\ \; & Cardinality(Distributor\cup IoT) \end{array}$$
$$\begin{array}{cc} NoUnpaidFee & ≜\\ \; & \forall \;q\in \{d\in IoT:dState[d]=I\_Final\}:\\ \; & \;\;\;\;\;\;\wedge balance[q]=InitBalance-NUMPIECES*PIECEPRICE \end{array}$$

The TotalBalanceInvariance and NoUnpaidFee are the invariant which is true for all the states satisfying the condition. The first condition ensures that the balance of the system is consistent. If the IoT device has paid for a piece of a patch, the balance of the delivering Distributor increases by price of the piece. The second condition ensures that the Distributor who has delivered a piece of the patch, is paid for the delivery.

Model Simulation and Validation: We performed simulation on TLA+ model verification tool to check the correctness of the proposed system and the critical conditions. We ran the experiment on a Windows 10-based Intel i7 quad-core machine with 32 GB RAM. Table 2 shows the model checking results with the properties for three distributors and three IoT devices with four-piece patch update. The simulation process took 240s and visited 1,174,249 distinct states. The simulation completed without encountering any error except for violation of NoUnpaidFee. We discuss this attack in the next section.

### 8.2. Threat Analysis

Reward interception: Since an attacker has complete control over the communication channel, he/she can spoof the redeem transaction, establish a channel with the distributor, and try to withdraw the reward to himself. To prevent this, the invoice generated for the payment contains the public key of the receiver (distributor). The IoT device verifies the public key before making the payment.

Malformed Channel: An attacker can try to change the channel established between the distributor and the IoT devices, thereby, disrupting the exchange of the patch updates and incentives. The success of the attack depends on compromising the Bitcoin network as the channel becomes public once the funding transaction is confirmed on the blockchain. Thus, unless the Bitcoin network is compromised, the channel attributes cannot be altered.

Denial-of-service (DoS) Exhausting Resources: An attacker, acting as an honest distributor or as an honest IoT node, can try to waste resources of the participating entity. However, the proposed framework does not require high demanding tasks to be performed. The only attack it can be subjected to is to hash a message and produce a signature, which is not difficult to compute.

Denial-of-service (DoS) Blockchain-related attacks: Another way to prevent patching from being performed is to censor transactions that reach the blockchain network. Aside from the eclipse attack, which has been explained above, such an attack is typically referred to as a censorship attack. Again, such type of attacks requires the Bitcoin network to be compromised, which is difficult by design.

Compromising a firmware’s integrity: A malicious actor acting as a vendor may try to announce a malicious software update to the IoT devices. In the proposed framework, it is a costly attack as the malicious vendor must commit a bid per IoT node on the blockchain.

Software downgrade attack: An alternative to the above attack, a malicious actor may try to push a compromised version of the patch update. Again, in the proposed framework, it is a costly attack as the malicious vendor must commit a bid per IoT node on the blockchain.

Greedy Distributor: A distributor, acting greedily, may request payment for pieces not sent to the IoT devices. The proposed framework enables the IoT device to verify the availability of the piece. The IoT device can reject the payment request for the pieces not received by it.

Greedy IoT Node: An IoT node, acting greedily, can request distinct pieces from separate distributors without releasing the payment. The attack can succeed in case the number of pieces of patch update is less than the number of distributors seeding the update. Under reasonable circumstances, we can assume that the number of distributors will be less than the number of pieces of the patch update (a 1MB file is split into 64 pieces). Also, in case the number of distributors are comparable to the number of pieces, the IoT device can cheat utmost once, after which the misbehavior will be detected and the IoT node will be blocked.

### 8.3. Properties

In this section, we will prove the properties of the protocol concerning the security goals in Section 3.3:Fair exchange: The protocol construction is based on first delivering the piece from the distributor to the IoT device and, in return, receiving the signature assuring the reward claim via the lightning network. In the protocol, the distributor can be listed on the DSN by first downloading the patch and hashing the file into an address, meaning that the distributor holds the patch binaries ready to deliver. Each piece in the torrent protocol is hashed and can be verified by the receiver. In the case of delivering false pieces from the distributor to the IoT device, the IoT will validate the piece against the hash and confirm its validity.The second phase is IoT signing the requested message. In this construction, the distributor can validate the signature only after receiving the signature, thus, putting himself in risk and will be compromising a fraction of the reward once per IoT lifetime. Since this phase starts only after validating the IoT public address in the handshake phase, the distributor can hold a blacklist of IoT, making it protected the next time the IoT requests a piece. In the transitional construction of the gradual release, both parties should have the same computational power to eliminate the ability of one party to abort between rounds and brute force the remaining information In this protocol, the availability of the piece is from multiple sources, eliminating the incentive of one party to redraw from the transfer.Patch integrity: The patch update file hash is listed in the BitTorrent protocol header. This, along with the use of a collision-free hash function patch, will lose its integrity with insignificant probability.Patch availability: In P${}^{4}$UIoT, the initial seed of the vendor is used throughout the process. The decentralized storage and Bitcoin network with the seed mentioned above ensures that at least one distributor can complete the file download, and thus, further ensures the availability of the patch update.

### 8.4. Execution Results

To evaluate the performance of the proposed framework, an experimental setup is created on the *regtest* network of Bitcoin using lxd containers. The *regtest* is configured to mimic the Bitcoin mainnet by adjusting the mining delay. Also, to simplify the scenario, a hub-spoke configuration is established, where the hub is running a full node, while all other entities run a light client connected to the hub. The containers are created on an 8 core, 30 GB machine with 1 tb disk space.

The experiment consists of different scenarios with a varying number of IoT devices and the size of the patch update. The number of vendors and distributors is fixed to understand the scalability of the network. The number of IoT devices are varied as 1, 10, 25, and 50. The patch size is varied as 10 kb, 100 kb, 1 mb, 10 mb, and 100 mb. The latency of patch delivery and the cost (transaction cost and fee) involved in delivering the patch update to every IoT device is recorded and presented in Table 3 and Table 4. The patch is split into equal size parts as per the BitTorrent protocol. Also, for the sake of simplicity, we assume the piece price to be one milli satoshi. The negotiation for the price of a piece can be performed offline and is beyond the scope of this paper. Figure 5 presents the variation of latency against the number of IoT devices for patch size 10 kb, 100 kb, and 1 mb.

We can see from Figure 5 that the distribution latency is almost linear for small size patch, while it increases exponentially for larger file size. After establishing a payment channel, the payment between two parties happens in a peer-to-peer fashion. Thus, the only latency in payment of incentive is the network delay, which in turn depends on the length of the lightning channel between the IoT device and the distributor. Also, the lightning network adds pseudo-path to the route to protect the privacy of the participants. In a real-world scenario, with more distributors participating in the process, the distribution latency will reduce even further with the distributor being near the IoT device.

Table 5 presents a monetary comparison between the proposed solution and IoTPatchPool, which is based on a blockchain-based patch delivery framework. We can observe from the table that the cost of channel establishment is significantly less compared to the contract deployment. Also, the contract deployment needs to be done for each patch update, while the channel establishment needs to be done only once and can be used until the commitment exhausts. Regarding the incentive for the patch delivery, the fees involved in the lightning network consists of two components, a standard base fees, which is roughly one satoshi, and liquidity fees, which is roughly 1% of the transacted amount. Considering the micropayment use case, the overall cost of the patch delivery will be significantly less compared to any pure blockchain-based solution.

## 9. Conclusions

We described and implemented P${}^{4}$UIoT, a pay-per-piece patch update for IoT software updates using a distributed storage network. The proposal combines a distributed file-sharing network and lightning network-based payment channels to transfer patch updates to the IoT devices. The lightning channel enables quick micropayments with minimum delay and zero transaction fees. A fair exchange of patch updates and micropayments is established gradually using a pay-per-piece exchange protocol. The privacy of the framework is maintained by the onion-routing of the lightning network. Thus, the information flows between the participating entities. We presented a formal specification of the proposed approach using the TLA+ formal language and checked the correctness of the model using the TLC model checker. We evaluated the resilience of the proposed framework against known threats such as reward interception, malformed channel, Denial-of-service (DoS) redeem with false message, exhausting resources, Blockchain-related attacks, compromising a firmware’s integrity, software downgrade attack, greedy distributor, and greedy IoT node. Also, the distribution latency and financial analysis of the framework is experimented, and the scalability of the framework is evaluated. As future work, formal security analysis can be conducted using tools like Scyther. Also, the statistical and cost analysis of the framework in a real-time scenario can be conducted to evaluate production readiness.

## Figures and Tables

**Figure 1 sensors-20-02156-f001:**
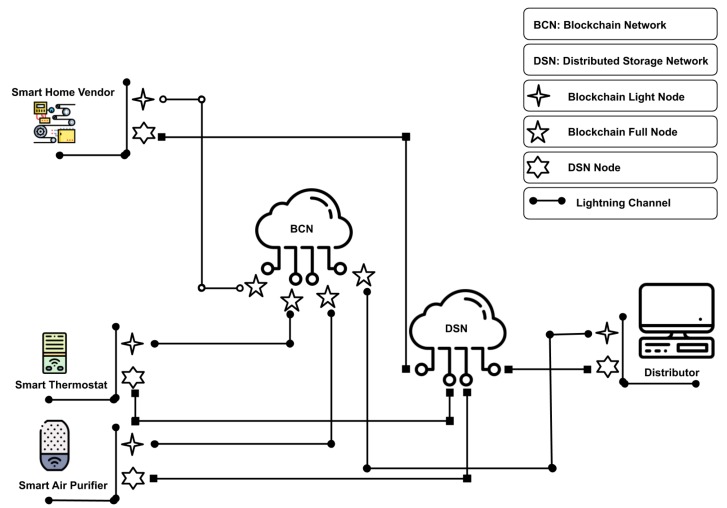
P${}^{4}$UIoT High level architecture overview.

**Figure 2 sensors-20-02156-f002:**
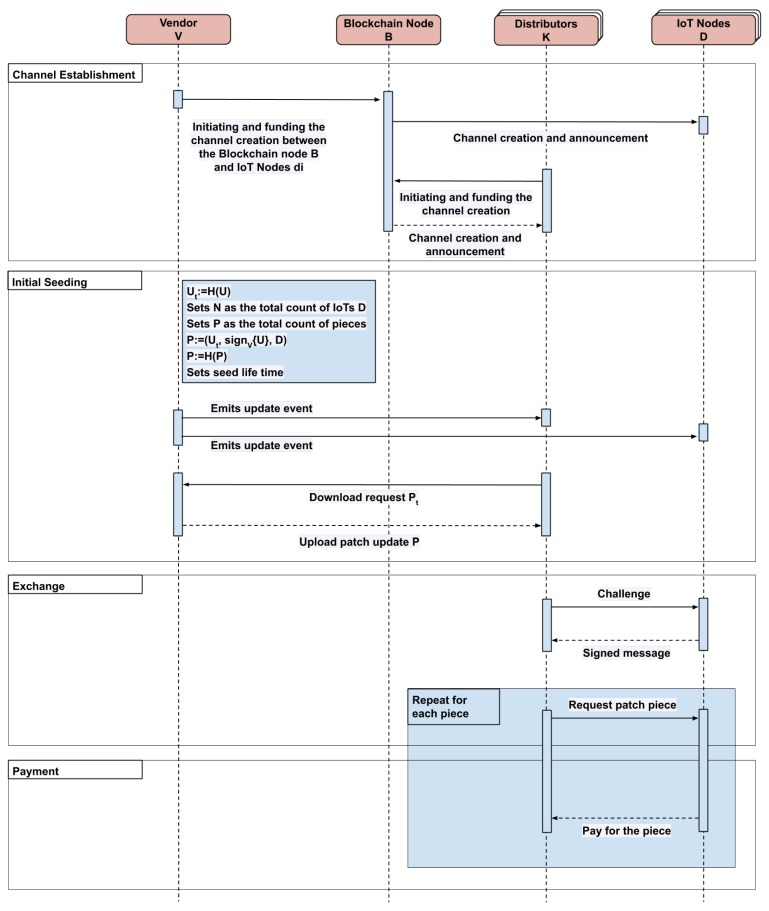
P${}^{4}$UIoT sequence diagram.

**Figure 3 sensors-20-02156-f003:**
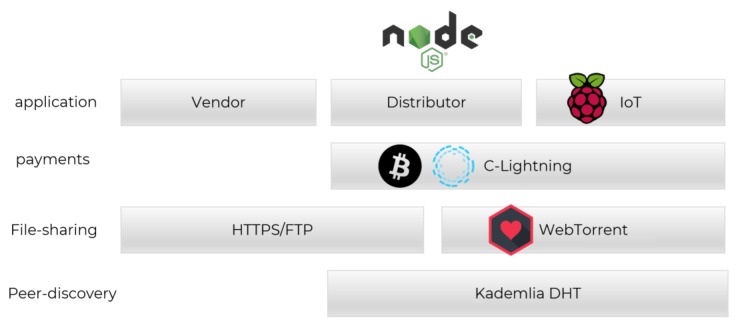
The technology stack used in P${}^{4}$UIoT implementation.

**Figure 4 sensors-20-02156-f004:**
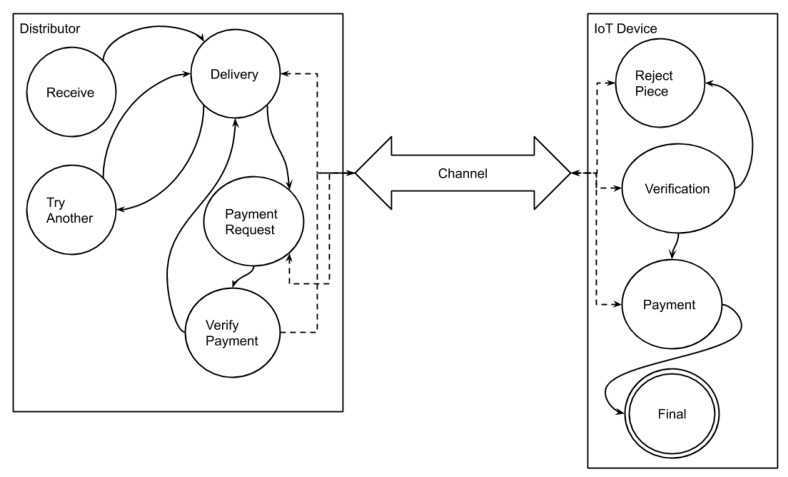
P${}^{4}$UIoT state transition diagram.

**Figure 5 sensors-20-02156-f005:**
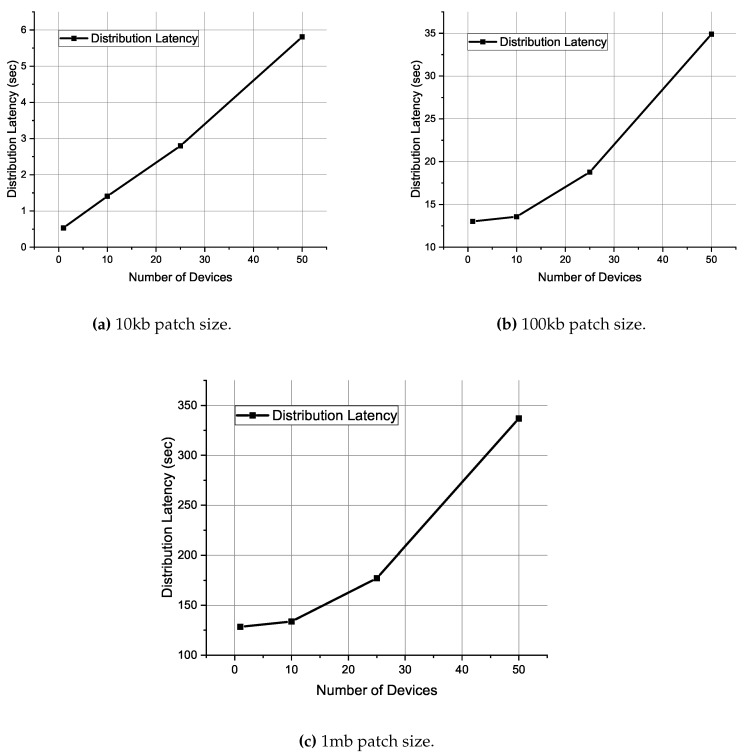
Distribution latency for patch size (**a**) 10 kb, (**b**) 100 kb, and (**c**) 1 mb patch size.

**Table 1 sensors-20-02156-t001:** Summary of related works (in the Solution column, D stands for design, I stands for implementation, and S stands for simulation).

Reference	Type	Distributed/Centralized	Incentives	Trust (Fair Exchange)	Scaling	Solution D, I, or S	Comment
Yu et al. (2015) [35]	IoT Patching	Centralized	–	–	–	D + I	Centralized IoT security via antivirus and patching.
Onuma et al. (2017) [36]	General	Centralized	–	–	–	D + I + S	Reduction in update latency using divide and distribute.
Huth et al. (2016) [37]	IoT Patching	Centralized	–	–	–	D + I + S	Secure software patch update in IoT using Physically Unclonable Functions.
Kim et al. (2017) [38]	IoT Patching	Centralized	–	–	–	D + I + S	Secure software patch delivery using Cloud.
Lee et al. (2017) [27]	IoT Patching	Distributed (Bitcoin Network)	–	–	–	D	distributed storage network and blockchain-based patch delivery framework.
Boudguiga et al. (2017) [28]	IoT Patching	Distributed (Multichain Network)	–	–	–	D + I	distributed storage network and blockchain-based patch delivery framework with trusted verifier.
Lee et al. (2018) [29]	IoT Patching	Distributed (Ethereum Network)	Eth	–	–	D + I + S	distributed storage network and blockchain-based incentivized patch delivery framework.
Popov (2017) [39]	General	Distributed (DAG Tangle)	IOTA	–	X	D + I + S	Incentivized distributed storage network.
Cohen (2003) [22]	General	Distributed (BitTorrent Network)	–	–	X	D + I + S	Incentivized distributed storage network with fair exchange.
Siacoin (2014) [43]	General	Distributed (Sia Network)	Siacoin	X	X	D + I + S	Incentivized distributed storage network.
Storj (2014) [44]	General	Distributed (Storj Network)	STORJ	–	X	D + I + S	Incentivized distributed storage network.
Swarm (2016) [45]	General	Distributed (Ethereum Network)	Eth	–	X	D	Incentivized distributed storage network.
Filecoin (2017) [46]	General	Distributed (Filecoin Network)	Filecoin	–	X	D + I	Incentivized distributed storage network.
Leiba et al. (2019) [30]	IoT Patching	Distributed (Ethereum Network & BitTorrent Network)	Eth	X	–	D + I + S	Blockchain-based incentivized distributed storage network with fair exchange.
P${}^{4}$UIoT	IoT Patching	Distributed (Bitcoin Network & BitTorrent Network) BitTorrent Network)	Bitcoin	X	X	D + I + S	Blockchain-based incentivized distributed storage network with fair exchange and scalability.

**Table 2 sensors-20-02156-t002:** Model checking results with three distributors and three iot devices with four pieces of patch file.

Time	Depth	States Found	Distinct States	Errors
780’	71	17,315,287	3,302,038	0

**Table 3 sensors-20-02156-t003:** Performance measure of distribution latency (sec).

Number of IoT Devices	Patch Size		
	10 kb	100 kb	1 mb
1	0.532	13.027	128.513
10	1.41	13.56	133.75
25	2.8	18.77	177.12
50	5.81	34.88	336.94

**Table 4 sensors-20-02156-t004:** Cost measure (millisatoshi).

Number of IoT Devices	Patch Size		
	10 kb	100 kb	1 mb
1	1	7	64
10	10	70	640
25	25	175	1600
50	50	350	3200

**Table 5 sensors-20-02156-t005:** Cost (in $) analysis and comparison with IoTPatchPool.

Transaction Type	Usage Frequency	Proposed Solution ($)	IoTPatchPool ($)
Create factory	per framework setup	–	0.36
Create contract	per update file	–	0.29
Create channel	per IoT device	0.017	–
Commit	per delivery	–	0.08
Reveal	per delivery	–	0.02
Delivery cost	per piece	0.000097 + 1% of transaction amount	–

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
