# Peer review of "P4UIoT: Pay-Per-Piece Patch Update Delivery for IoT Using Gradual Release"

_sensors, 2020, doi:10.3390/s20072156_

Round 1

Reviewer 1 Report

This paper proposes a patch distribution network for IoT devices which utilizes a pay-per-piece exchange protocol. The paper is well written and structured and the flow of the paper is nice. The experiment conducted by the authors provide some encouraging figures. One issue that the authors might need to elaborate is the identification where the authors propose the use of public key cryptography which for some IoT node might be the ideal solution due to resource constraints.
Some minor English syntax errors have to be corrected.

Author Response

Q1.1: One issue that the authors might need to elaborate is the identification where the authors propose the use of public key cryptography which for some IoT node might be the ideal solution due to resource constraints. 

A1.1: We agree with reviewer’s comment regarding the limitation of public key cryptography in resource constrained devices. In the article, we described Schnorr Signatures as a possible signature aggregation scheme used in the bitcoin network. The scheme takes O(n/log(n)) time for verifying n signatures. Also, bitcoin is based on ECDSA which when compared to RSA reduces the key length significantly. This in turn improves the verification time. We have added the following paragraph (page 10 line 306):

“The Schnorr Signatures and ECDSA together can improve the verification time with signature aggregation and key size reduction [53]. This is particularly advantageous in the case of resource-constrained devices.”

  1. Imem, A.A. Comparison and evaluation of digital signature schemes employed in NDN network. arXiv preprint arXiv:1508.00184 2015.

Q1.2: Some minor English syntax errors have to be corrected.

A1.2: We thank the reviewer for the comment. We carefully re-read the paper and removed all the typos from the revised manuscript.

Reviewer 2 Report

This work described and implemented P4UIoT, a pay-per-piece patch update for IoT software updates using distributed storage network. The proposal combines a distributed file sharing network and lightning network based payment channels to transfer patch update to the IoT devices.

Major issues:

- This work uses bitcoin regtest network, but it is unclear the consensus mechanism and any update / modification was applied.

- This work mainly compares with IoTPatchPool, but there are some other similar work can should be considered:

Patch Transporter: Incentivized, Decentralized Software Patch System for WSN and IoT Environments. Sensors 18(2): 574 (2018)

- 7.1. Threat Analysis: this section is not helpful, as there are no formal analysis presented.

- In addition, there are no results under attacks. It is expected some practical performance should be given under various attacks mentioned in this work.

Minor issues:

- "5.1. Introduction" -> this section title should be more concrete

- Writing styles should be checked, e.g., "We also assumes" -> assume

Author Response

Q2.1: This work uses bitcoin regtest network, but it is unclear the consensus mechanism and any update / modification was applied.

A2.1: We thank the reviewer for the comment. The proposed solution uses the bitcoin and the lightning network as is without any changes. We agree with the reviewer that this is important information and, thus, we have included this on page 12 line 383 as follows:

“The bitcoin and lightning network protocols are used as is and no changes

has been made to the original protocols.”

Q2.2: This work mainly compares with IoTPatchPool, but there are some other similar work can should be considered:

Patch Transporter: Incentivized, Decentralized Software Patch System for WSN and IoT Environments. Sensors 18(2): 574 (2018)

A2.2: We agree with the reviewer regarding comparing other solutions with the proposed approach. However, we want to point out that the IoTPatchPool essentially represents a general blockchain based distribution solution, where the blockchain acts as a trusted party ensuring security guarantees. The proposed solution solves the latency and scalability issues with this general solution. 

In addition, we include the Patch Transporter paper in the related section on page 3 line 121 and we compare it to our proposed framework:

“Lee [29] proposed a blockchain-based solution for patch distribution with the incentive for delivery. A unique receipt is generated for each delivered patch, and only the distributor successfully providing the patch can claim the reward. This is achieved by registering the identity of the distributor (encryption keys), which is then used to preserve the integrity of the patch update. However, the uniqueness property forces the vendor to distribute the package to every distributor to have fairness in the system. Thus, the system has similar bandwidth requirements as those of a centralized solution. Furthermore, the creation of blockchain transactions for each patch delivery by a distributor is a costly and time-consuming process.  Also, the solution requires the IoT device to maintain a digital wallet to pay for the update, which introduces additional vulnerabilities and attack vectors. Finally, the wallets need to be provisioned, causing additional transactions on the blockchain.”

Q2.3: 7.1. Threat Analysis: this section is not helpful, as there are no formal analysis presented.

A2.3: We agree with the reviewer that a formal analysis would further strengthen the manuscript. We performed a formal analysis of the proposed scheme based on the paper suggested by the reviewer. We added Section 7 outlining the formal specification of the proposed approach and presented the simulation result in Section 8. 

Q2.4: In addition, there are no results under attacks. It is expected some practical performance should be given under various attacks mentioned in this work.

A2.4: We understand the reviewer’s observation and thus, we added the simulation results evaluating the correctness of the model based on the formal specification. However, we want to point out that the threat analysis presented earlier in Section 8 assumed the security guarantees of the underlying platform (Bitcoin and lightning network) and thus, logically evaluated the security guarantees of the proposed scheme. We, thus, kept the section related to threat analysis.  

Q2.5: "5.1. Introduction" -> this section title should be more concrete

A2.5: We combined the section with the introductory section. We thank the reviewer for the comment.  

Q2.6: Writing styles should be checked, e.g., "We also assumes" -> assume

A2.6: We thank the reviewer for the comment. We carefully re-read the paper and removed all the typos from the revised manuscript.

Reviewer 3 Report

The paper is very interesting and contributes to the state of the art of IoT-oriented methodologies for applying patches in case of security updates required to applied. Following a P2P approach, the idea proposed by authors is able to timely reach the devices and contributes to enhance the overall security of the interested architecture.

Authors are required to carefully check their paper, in order to correct possible typos contained in the current version of the paper, as an example:

  • extended definition of P4UIoT in the abstract should be compliant with the title of the paper
  • "p2p" should be "P2P"
  • "logn" should be "log(n)"
  • please verify for additional blank spaces not required in mathematical equations
  • please verify in Fig. 3 WebTorrent and Kademlia, which are underlined

Author Response

Q3.1: Authors are required to carefully check their paper, in order to correct possible typos contained in the current version of the paper, as an example:

  • extended definition of P4UIoT in the abstract should be compliant with the title of the paper
  • "p2p" should be "P2P"
  • "logn" should be "log(n)"
  • please verify for additional blank spaces not required in mathematical equations
  • please verify in Fig. 3 WebTorrent and Kademlia, which are underlined

A3.1: We thank the reviewer for the comment. We carefully re-read the paper and removed all the typos from the revised manuscript. Also, we addressed the specific improvements suggested by the reviewer in the revised manuscript.

Round 2

Reviewer 2 Report

I have no further comments.